# OTU7B Modulates the Mosquito Immune Response to *Beauveria bassiana* Infection via Deubiquitination of the Toll Adaptor TRAF4

Yanhong Wang,[a,b] Mengmeng Chang,[a,b] Mao Wang,[a,b] Yannan Ji,[a,b] Xiaomei Sun,[a,b] Alexander S. Raikhel,[c] ⓘ Zhen Zou[a,b]

aState Key Laboratory of Integrated Management of Pest Insects and Rodents, Institute of Zoology, Chinese Academy of Sciences, Beijing, China
bCAS Center for Excellence in Biotic Interactions, University of Chinese Academy of Sciences, Beijing, China
cDepartment of Entomology and Institute for Integrative Genome Biology, University of California, Riverside, California, USA

Yanhong Wang, Mengmeng Chang, and Mao Wang contributed equally to this work. Author order was determined in order of decreasing seniority.

**ABSTRACT** The *Aedes aegypti* mosquito transmits devastating flaviviruses, such as Zika, dengue, and yellow fever viruses. For more effective control of the vector, the pathogenicity of *Beauveria bassiana*, a fungus commonly used for biological control of pest insects, may be enhanced based on in-depth knowledge of molecular interactions between the pathogen and its host. Here, we identified a mechanism employed by *B. bassiana*, which efficiently blocks the *Ae. aegypti* antifungal immune response by a protease that contains an ovarian tumor (OTU) domain. RNA-sequencing analysis showed that the depletion of OTU7B significantly upregulates the mRNA level of immunity-related genes after a challenge of the fungus. CRISPR-Cas9 knockout of OTU7B conferred a higher resistance of mosquitoes to the fungus *B. bassiana*. OTU7B suppressed activation of the immune response by preventing nuclear translocation of the NF-$\kappa$B transcription factor Rel1, a mosquito orthologue of *Drosophila* Dorsal. Further studies identified tumor necrosis factor receptor-associated factor 4 (TRAF4) as an interacting protein of OTU7B. TRAF4-deficient mosquitoes were more sensitive to fungal infection, indicating TRAF4 to be the adaptor protein that activates the Toll pathway. TRAF4 is K63-link polyubiquitinated at K338 residue upon immune challenge. However, OTU7B inhibited the immune signaling by enzymatically removing the polyubiquitin chains of mosquito TRAF4. Thus, this study has uncovered a novel mechanism of fungal action against the host innate immunity, providing a platform for further improvement of fungal pathogen effectiveness.

**IMPORTANCE** Insects use innate immunity to defend against microbial infection. The Toll pathway is a major immune signaling pathway that is associated with the antifungal immune response in mosquitoes. Our study identified a fungal-induced deubiquitinase, OTU7B, which, when knocked out, promotes the translocation of the NF-$\kappa$B factor Rel1 into the nucleus and confers enhanced resistance to fungal infection. We further found the counterpart of OTU7B, TRAF4, which is a component of the Toll pathway and acts as an adaptor protein. OTU7B enzymatically removes K63-linked polyubiquitin chains from TRAF4. The immune response is suppressed, and mosquitoes become much more sensitive to the *Beauveria bassiana* infection. Our findings reveal a novel mechanism of fungal action against the host innate immunity.

**KEYWORDS** fungus, innate immunity, Toll pathway, deubiquitination, mosquito, host-microbe interaction

Address correspondence to Zhen Zou, zouzhen@ioz.ac.cn, or Alexander S. Raikhel, alexander.raikhel@ucr.edu.

The authors declare no conflict of interest.

The *Aedes aegypti* mosquito is the major vector of devastating arboviral diseases. Notably, about 400 million people become infected with just Dengue fever, with 40,000 deaths from severe complications from the disease, annually. With an anti-

Dengue vaccine still in development, control of mosquitoes is an important means of managing the disease. Several entomopathogenic fungi, such as *Beauveria bassiana* and *Metarhizium anisopliae*, are used against a wide range of insect pests, including mosquitoes (1, 2). However, invading fungal pathogens are attacked by the powerful innate immune system of insects, lowering the effectiveness of these fungi. To evade the attack of host immunity, microbes have adopted many strategies. For example, the lysin motif (LysM)-containing proteins help *B. bassiana* to escape the insect immune surveillance (3). MoCIDP4, an effector of the fungal pathogen *Magnaporthe oryzae*, exploits the mitochondrial dynamics to reduce plant immunity (4). *Fonsecaea monophora*, a pathogen of chromoblastomycosis, is able to bind to C-type lectin Mincle and modulate host antifungal responses (5). Further studies of mechanisms of antifungal immunity and fungal counterattacks are essential for the improvement of biocontrol fungal agents.

Previous studies have shown that the Toll pathway controls the antifungal immune response in flies and mosquitoes (6–8). In *Drosophila melanogaster*, the Toll pathway is mediated by an extracellular serine protease cascade in response to infection with fungi or Gram-positive bacteria (9). Through the Toll receptor, the immune signal is transmitted by an intracellular signal cascade composed of MyD88, Tube, and Pelle, leading to the degradation of Cactus. The latter event causes the release of the NF-$\kappa$B transcription factors Dorsal and Dif and their nuclear translocation. In the nucleus, these two transcription factors activate the expression of an array of immune effectors, such as antimicrobial peptides (AMPs) (9, 10). *Ae. aegypti* Rel1 (AaRel1), an orthologue of Dorsal, is a key molecule in the Toll immune pathway (11). In *Ae. aegypti*, fungal infection activates the Toll pathway, causing Rel1 translocation into the nucleus, causing the expression of a lot of immune effectors, such as AMPs and lysozymes (8).

Ubiquitination is considered one of the key signaling events in the regulation of innate immune signaling pathways, particularly NF-$\kappa$B pathways in mammals and insects (12–16). A ubiquitin molecule has 76 amino acid residues, including 7 internal lysines (Lys): K6, K11, K27, K29, K33, K48, and K63. The process of ubiquitination requires the coordination of three kinds of enzymes: ubiquitin-activating enzyme E1, ubiquitin-conjugating enzyme E2, and ubiquitin-ligase E3, wherein E3 can be removed by deubiquitinases (DUBs). E3 ligase links the ubiquitin to Lys residues of a target protein and that determines the specificity of ubiquitination (17). During a ubiquitination reaction, single or multiple ubiquitins are linked with target proteins (18). In the process of polyubiquitination, after the first ubiquitin is linked to the target protein, other ubiquitins attach to different Lys residues of the previous ubiquitin molecule to form different types of polyubiquitin chains that determine the fate of target proteins. The clearest studied are the K48- and K63-linked polyubiquitin chains. The former facilitates proteasomal degradation of proteins, and the latter mainly mediates protein interactions during signal transduction (19, 20).

In mammals, K63-linked polyubiquitination of receptor-interacting proteins, including RIP1, tumor necrosis factor receptor-associated factor 2 and 6 (TRAF2 and TRAF6), and the NF-$\kappa$B modulator protein NEMO, is involved in the stimulation of transforming growth factor $\beta$-activated kinase 1 (TAK1) and I$\kappa$B kinase (IKK). These K63-linked polyubiquitin chains act as scaffolds for the TAK1 protein kinase complexes (21). In *D. melanogaster*, immune deficiency (Imd) is K63-linked polyubiquitinated upon Gram-negative bacterial infection, activating TAK1, which triggers the removal of K63-linked chains from Imd and the addition of K48-linked chains to the molecule. K48-linked polyubiquitination mediates the proteasomal degradation of Imd, as it possibly restores homeostasis to the immune response (22).

Here, we identified a novel mechanism by which the fungus *B. bassiana* affects mosquito innate immunity. We have shown that fungal infection triggers ovarian tumor (OTU) domain-containing protease (OTU7B) expression, which in turn removes the K63-linked polyubiquitin chains of TRAF4. As a result, the nuclear translocation of the

NF-$\kappa$B transcription factor Rel1 is blocked. Our findings provide the potential for the development of more efficient entomopathogenic fungi, such as *B. bassiana*.

## RESULTS

**Fungal-induced OTU7B suppresses mosquito immunity.** OTU-domain-containing DUBs are involved in the regulation of mammalian innate immunity (23–25). However, little is known about this class of enzymes in insects, including mosquitoes. We have identified seven OTUs in the *Ae. aegypti* genome and investigated their responses after septic injections with conidia of *B. bassiana* for 24 h and 48 h. Quantitative real-time PCR (qRT-PCR) results revealed that the mRNA levels of four OTUs (OTU5B, Trabid, OTU, and Otubain-like) were suppressed after fungal infection; transcripts of two OTUs did not change (Fig. S1A in the supplemental material). Only OTU7B, which contains an OTU domain at the N terminal and two A20-type zinc fingers at the C terminal, was highly upregulated both at mRNA level and protein level postinfection (Fig. 1A). This finding coincides with the results of immunofluorescence analysis showing that the OTU7B signal in fat body cells increased significantly after fungal infection (Fig. 1B). Next, we studied mosquito survival after RNA interference (RNAi) of OTUs. Three days after injection of double-stranded RNA (dsRNA), mosquitoes were infected with *B. bassiana*, and their survival rates were counted. The results showed that after OTU7B dsRNA injection, the survival rate significantly increased. However, there was minimal change in survival rates of mosquitoes injected with dsRNAs of other OTUs relative to the control (knockdown of enhanced green fluorescent protein, iEGFP) after *B. bassiana* infection (Fig. S1B). Thus, RNAi silencing of OTU7B leads to increased resistance to fungal infection. This suggests that *B. bassiana* may manipulate host immunity by inducing a high expression of OTU7B.

Previous studies have shown that *Ae. aegypti* Rel1A, which is an orthologue of *Drosophila* Dorsal, was involved in antifungal immune response, and Rel1B acted cooperatively with Rel1A to enhance immune activation (11). To understand the regulatory mechanism of OTU7B expression induced by fungus, we identified one NF-$\kappa$B binding motif (GGGAACCTAT) in the promoter of OTU7B. Then, dual-luciferase reporter assays were conducted to confirm whether *OTU7B* was directly regulated by Rel1. The luciferase activity significantly increased in Rel1A-transfected Aag2 cells (Fig. 1C). We further confirmed that Rel1A interacts with the NF-$\kappa$B binding motif in *OTU7B* upstream by means of electrophoretic mobility shift assay (EMSA). Results showed that a retardation band was colocalized with a biotin-labeled probe, and this band reduced when the concentration of Rel1A decreases or after the addition of a 50-fold excess of unlabeled probe (Fig. S1C).

To investigate the role of OTU7B further, we performed RNA sequencing (RNA-seq) analysis linked with OTU7B RNAi to study the effect of OTU7B on the mRNA level of immunity-related genes. For this assay, 1 $\mu$g OTU7B dsRNA was injected into mosquitoes (iOTU7B); 3 days later, mosquitoes were infected with fungus conidia for 48 h (iOTU7B-Bb); and then, fat body samples were collected for RNA extraction. iEGFP mosquitoes served as a control. Hierarchical clustering analysis of differentially expressed immunity-related genes (DEGs; $P \leq 0.05$) revealed that numerous cohorts were upregulated in iOTU7B-Bb (Fig. S1D; Table S1). We next selected upregulated (fold change $\geq 1.5$) immunity-related gene cohorts from iEGFP-Bb (75 genes), iOTU7B-PBS (66 genes), and iOTU7B-Bb (148 genes) for further analysis. We created a Venn diagram that showed that 46 immunity-related genes were commonly upregulated in all three treatment groups. These genes belong to the functional categories of recognition, signaling, and effectors. Most of these were upregulated in iOTU7B-Bb (Fig. 1D; Table S2), suggesting that OTU7B elicits a strong inhibitory effect on the immune response. Besides, when the function of OTU7B was abolished by RNAi, fungal infection boosted the expression of genes encoding immune effectors, including AMPs, thioester-containing proteins (TEPs), and pro-phenoloxidases (PPOs), while several heme-peroxidases (HPXs) were downregulated (Fig. S1E). Thus, it appears that OTU7B plays an essential role in

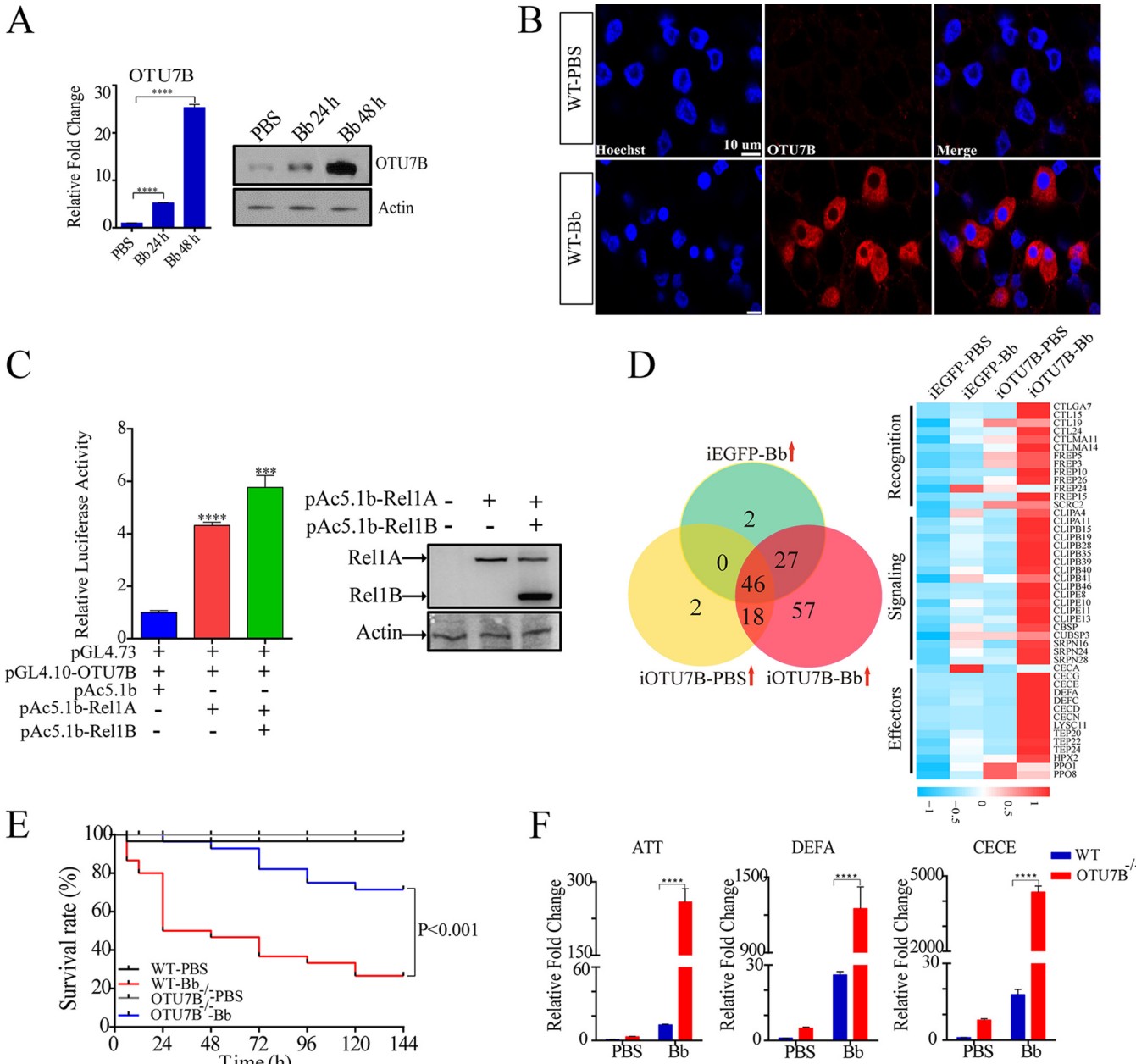

**FIG 1** OTU7B responses to fungal infection and suppression of the mosquito immune system. (A) qRT-PCR and immunoblotting analysis of the expression of OTU7B in mosquitoes. (B) Immunofluorescence staining shows the level of OTU7B (red) in fat bodies. WT-PBS, wild-type mosquitoes treated with PBS; WT-Bb, wild-type mosquitoes infected with fungi for 48 h. Scale bar, 10 $\mu$m. (C) The transcriptional activation of *OTU7B* promoter in Rel1A-overexpressed Aag2 cells. Immunoblotting was used to detect expression of Rel1A/Rel1B using anti-V5 antibody. Anti-actin antibody was used as the control. (D) Transcriptome results show the differential expression of immunity-related gene cohorts. (Left) Venn diagram indicating upregulated immunity-related genes (1.5-fold) in different treatment groups. (Right) Gene expression profiles of immunity-related genes that are commonly upregulated in iEGFP-Bb, iOTU7B-PBS, and iOTU7B-Bb. (E) Survival assays revealed the role of OTU7B in mosquito antifungal immunity. The OTU7B$^{-/-}$ is more resistant to *B. bassiana* infection than the wild-type (WT) ($P < 0.001$). (F) qRT-PCR results showed the transcription levels of selected immune effectors in WT and OTU7B mutant (OTU7B$^{-/-}$) mosquitoes treated with PBS or fungi (Bb). The data are shown as mean $\pm$ SEM. The data were normalized to the levels of WT-PBS. ****, $P < 0.0001$; the Student's *t* test was used for statistical analysis. The experiments were repeated three times.

mosquito antifungal immunity by preventing the automatic transcriptional activation of downstream immune effectors.

To better describe the function of OTU7B, we generated OTU7B loss-of-function (OTU7B$^{-/-}$) mosquitoes by means of CRISPR-Cas9. Pairwise alignment of wild-type and OTU7B$^{-/-}$ genotypes at a sgRNA-targeted genomic region showed that the OTU7B$^{-/-}$ line had a 4-bp deletion and an 11-bp insertion in the first exon. Immunoblotting confirmed that OTU7B was successfully knocked out (Fig. S2A). In addition, we found that

OTU7B$^{-/-}$ mosquitoes had a smaller body size and lighter body weight (Fig. S2B). Furthermore, they showed a smaller lipid droplet size and a lower level of triacylglycerol (Fig. S2C). Transcriptome results also demonstrated that the mRNA level of genes involved in lipid metabolism increased after OTU7B knockdown (Fig. S2D), consistent with the notion that immune response is an energy-consuming process (26). Moreover, several immunity-related genes, particularly AMP genes, were significantly upregulated, suggesting that the immune response was activated (Fig. S1D and E).

Next, we analyzed the survival rate of OTU7B$^{-/-}$ mosquitoes following fungal infection. Compared with wild-type, their survival rate significantly increased after fungal challenge, suggesting a critical role of OTU7B in antifungal immunity of mosquitoes (Fig. 1E). Based on the results of transcriptome analysis, we selected three upregulated AMP genes (*ATT*, *DEFA*, and *CECE*) and studied their expression in OTU7B$^{-/-}$ mosquitoes. qRT-PCR results showed that the mRNA levels of these AMPs were induced in both wild-type and OTU7B$^{-/-}$ mosquitoes after fungal infection. However, much higher induction levels of *ATT*, *DEFA*, and *CECE* were observed in OTU7B$^{-/-}$ mosquitoes, increasing by about 200, 1,200, and 4,000-fold, respectively (Fig. 1F). Taken together, these results suggest that OTU7B is highly expressed after fungal infection and plays a role in suppression immunity.

**OTU7B negatively regulates mosquito antifungal immune response by blocking the nuclear translocation of Rel1.** To explore whether OTU7B controls the immune response of mosquitoes by restricting Rel1, we conducted the immunostaining assay. The results showed that the Rel1 signal was hardly visible in fat body cells of wild-type and OTU7B$^{-/-}$ mosquitoes treated with PBS. However, after fungal infection, Rel1 immunosignal was detected in both types of mosquitoes. However, a highly strong immunosignal was observed in nuclei of fat body cells of OTU7B$^{-/-}$ mosquitoes (Fig. 2A). Immunoblotting provided a second detection method monitoring Rel1 nuclear translocation. For that, nucleoproteins from fat bodies of both wild-type and OTU7B$^{-/-}$ mosquitoes were extracted to measure the amount of Rel1 protein after different treatments. Immunoblotting results showed the presence of a weak Rel1 band in wild-type and OTU7B$^{-/-}$ mosquitoes after PBS infection. After fungal infection, the Rel1 protein band was considerably stronger in nucleoproteins of wild-type mosquitoes, but an even more robust band was identified in OTU7B$^{-/-}$ mosquitoes (Fig. 2B). These results suggest that OTU7B likely prevents Rel1 from nuclear entry.

Next, we studied the interrelation of OTU7B and Rel1 by RNAi. Survival assays showed that after Rel1 dsRNA injection (iRel1), survival rates significantly decreased in both wild-type and OTU7B$^{-/-}$ mosquitoes after fungal infection (Fig. 2C). This indicates that although fungal resistance of mosquitoes was enhanced after OTU7B knockout, it decreased again after Rel1 knockdown. Consistently, in Rel1 RNAi mosquitoes, the mRNA levels of *DEFA* and *CECE* significantly decreased after fungal infection in both wild-type and OTU7B$^{-/-}$ mosquitoes (Fig. 2D). Taken together, these results suggest that fungal infection enables OTU7B to control the transcription of immune effectors such as AMPs by restricting the nuclear translocation of Rel1.

**OTU7B interacts with the Toll adapter TRAF4 after fungal infection.** To further understand how OTU7B inhibits the antifungal immune response of mosquitoes, we searched for the potential interaction molecules of OTU7B. Aag2 cells were transfected with pAC5.1b-OTU7B, and 24 h later, Curdlan (a water-insoluble $\beta$-1,3-glucan) was added into cells at different time points. Immunoblotting showed that OTU7B was significantly upregulated at 6 h postapplication of Curdlan (Fig. S3A). The cell lysates were immunoprecipitated with anti-V5 antibody, and then the samples were separated by SDS-PAGE and analyzed by liquid chromatography-tandem mass spectrometry (LC-MS/MS) analysis (Fig. 3A). Based on *Ae. aegypti* protein database (https://vectorbase.org), some proteins (coverage >10%) identified from Curdlan-treated samples are shown in Fig. 3A, and the total list of proteins identified from the two samples are shown in Table S3 and S4 in the supplemental material. Previous studies have shown that TRAF proteins are key signaling molecules involved in the activation of various types of receptor-mediated signaling, such as the NF-$\kappa$B pathway and mitogen-

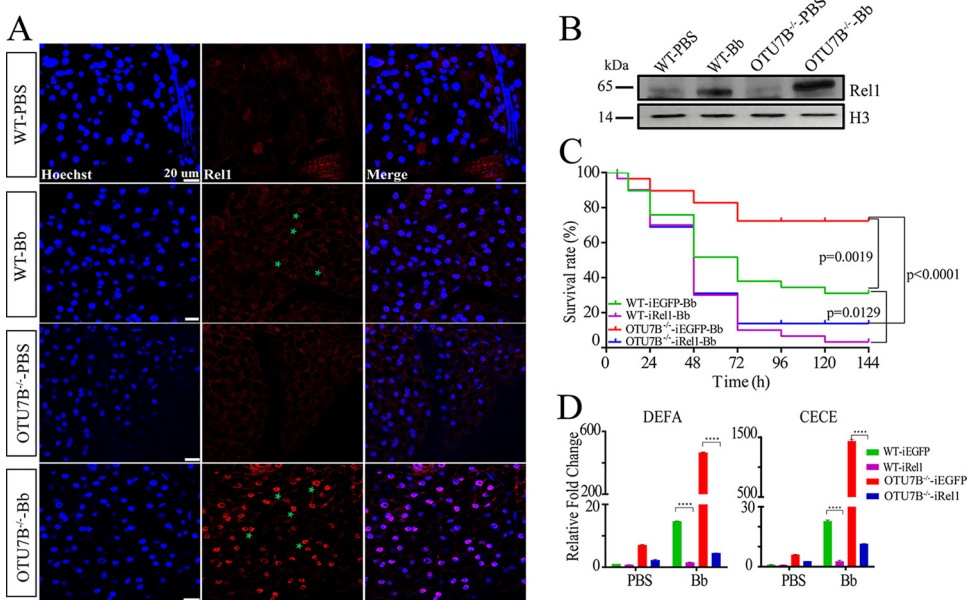

**FIG 2** OTU7B controls Rel1 entry into the nucleus after immune challenge. (A) Immunofluorescence staining shows the localization of Rel1 (red) in mosquito fat bodies. Scale bar, 20 $\mu$m. Green stars indicate Rel1 entering the nucleus. (B) Immunoblotting results show the expression of Rel1 in the nuclei of mosquitoes subjected to different treatments. Anti-histone3 (H3) antibody was used as loading control. WT-PBS or WT-Bb, wild-type mosquitoes treated with PBS or fungi; OTU7B$^{-/-}$-PBS or OTU7B$^{-/-}$-Bb, OTU7B mutant mosquitoes treated with PBS or fungi. (C) The survival rates rapidly decreased after treatment with Rel1 dsRNA. Each experiment was performed in triplicate. (D) The relative mRNA levels of *DEFA* and *CECE* in WT and OTU7B$^{-/-}$ mosquito that were treated with dsRNA. ****, $P < 0.0001$. The experiments were repeated three times. The data were subject to statistical analysis using the Student's *t* test. WT-iEGFP or WT-iRel1 and OTU7B$^{-/-}$-iEGFP or OTU7B$^{-/-}$-iRel1, wild-type or OTU7B mutant mosquito infection with fungi (Bb) or PBS after treatment with EGFP or Rel1 dsRNA.

activated protein kinase signaling cascades (27, 28). In *D. melanogaster*, dTRAF2, also known as dTRAF6, is a direct downstream target of Pelle and plays a role in the activation of the Toll signaling pathway (29, 30). Another *Drosophila* TRAF protein, dTRAF1, also known as dTRAF4, has been proven to interact with Pelle. Coexpression of Pelle and dTRAF1 in human cells significantly upregulates the NF-$\kappa$B activity (31). AaTRAF4 was identified from the Curdlan-treated cells, and our phylogenic analysis indicates it belongs to the same clade with dTRAF4 and dTRAF6 (Fig. S3B), and a glutathione *S* transferase pulldown (GST pulldown) assay determined the existence of His-tagged-TRAF4 (His-TRAF4) in the GST-tagged Pelle (GST-Pelle) immunoprecipitation, confirming their interaction (Fig. S3C). These results suggest that TRAF4 may play a similar role in the NF-$\kappa$B pathway in mosquitoes. Thus, we selected it as a potential counterpart of OTU7B (Fig. 3A).

To confirm the interaction between TRAF4 and OTU7B, we conducted coimmunoprecipitation assays using Aag2 cells that coexpressed HA-tagged TRAF4 and V5-tagged OTU7B. The results showed that when the V5-OTU7B fusion protein was immunoprecipitated from cell lysate with anti-V5 antibody, HA-TRAF4 was identified in the immunocomplex with anti-HA antibody (Fig. 3B, left). Similarly, when the HA-TRAF4 fusion protein was immunoprecipitated using anti-HA antibody, V5-OTU7B was detected in the immunocomplex using anti-V5 antibody (Fig. 3B, right). V5-OTU7B and HA-TRAF4 can be identified in total extract using anti-V5 and anti-HA antibodies, respectively (Fig. 3B, bottom). In addition, our immunostaining assays showed overlapping signals between endogenous OTU7B and TRAF4 in *Ae. aegypti* fat bodies after fungal treatment (Fig. 3C), while in OTU7B$^{-/-}$ mosquitoes, after fungal infection, the localization of TRAF4 is different from that of wild-type mosquitoes (Fig. S3D), indicating that OTU7B restricts the distribution of TRAF4 in the immune response. Thus, OTU7B interacts with the Toll adapter TRAF4 after fungal infection.

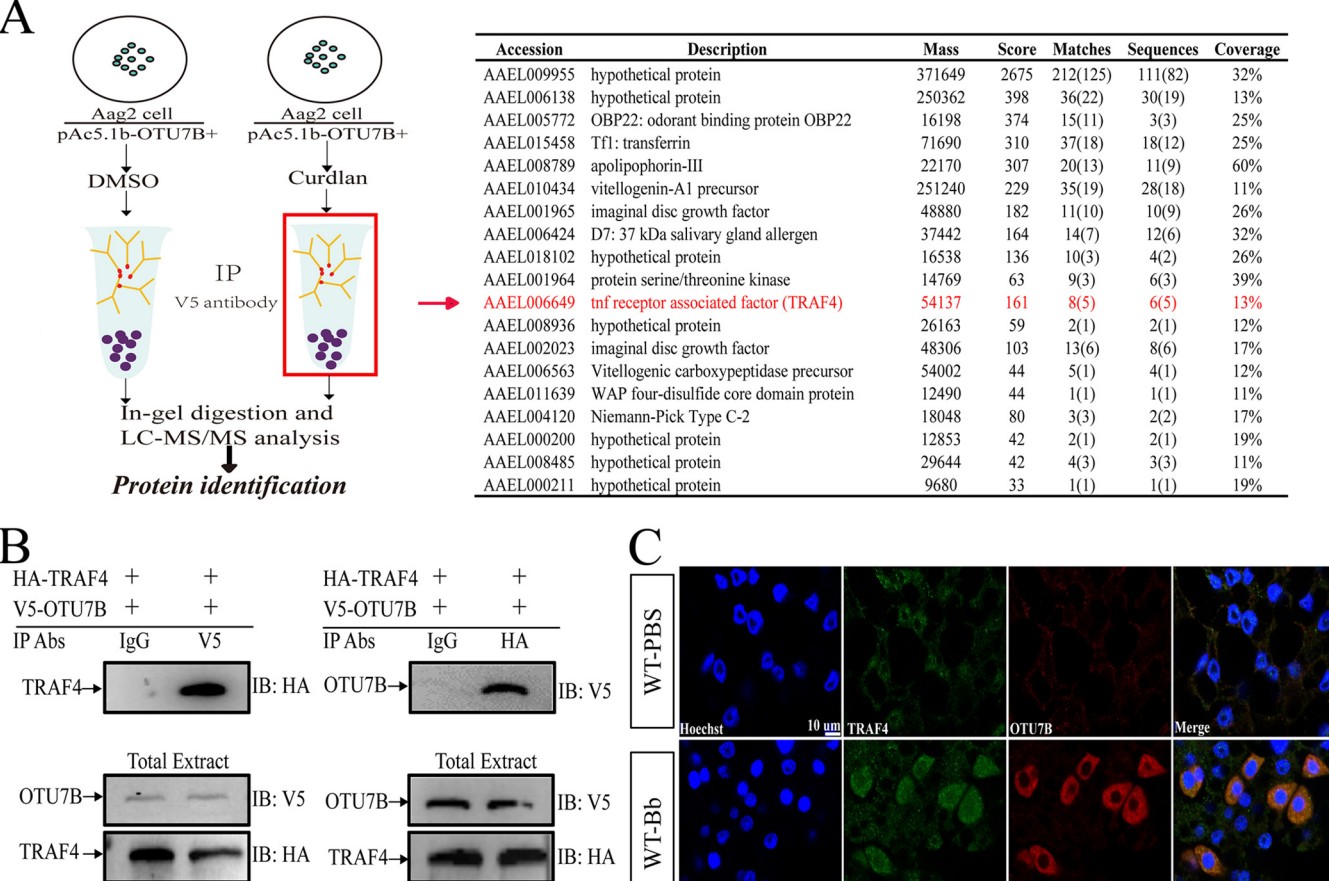

**FIG 3** Identification of TRAF4 as a protein that interacts with OTU7B after fungal infection. (A) Schematic procedure of immunoprecipitation and LC-MS/MS is shown, and the red mark implies the identified protein that potentially interacts with OTU7B after fungal infection. (B) Interaction of OTU7B and TRAF4 in mosquito cells. Aag2 cells were cotransfected with V5-OTU7B and HA-TRAF4 plasmids and then treated with Curdlan. Different antibodies used for immunoprecipitation (IP Abs) are indicated on top of the diagrams (IgG, control mouse IgG; V5, anti-V5 antibody; HA, anti-HA antibody), and the antibodies applied for immunoblotting (IB) are indicated on the right. Expression of exogenous in total extract is shown at the bottom. (C) Immunofluorescence showing that TRAF4 (green) and OTU7B (red) were coexpressed in the treated fat body cells. Scale bar, 10 $\mu$m.

**TRAF4 plays a critical role in antifungal immunity through the mosquito Toll pathway.** Because we have determined that OTU7B forms a complex with TRAF4 after fungal infection, we studied the role of TRAF4 in mosquito immunity. We first investigated the responses of TRAF4 to fungal infection. Immunoblotting assays showed that TRAF4 protein abundance was elevated after immune challenge (Fig. 4A). The mosquitoes with RNAi knockdown TRAF4 exhibited higher sensitivity to fungi (Fig. S4A), and the mRNA level of *DEFA* and *CECE* also decreased in iTRAF4 mosquitoes (Fig. S4B).

To further address the role of TRAF4 in immune responses, we used CRISPR-Cas9 to construct TRAF4-deficient mosquitoes, but no homozygous (TRAF4$^{-/-}$) mosquito was identified, and heterozygous lines (TRAF4$^{+/-}$) were used in subsequent experiments. Pairwise alignment assay showed that the *TRAF4* mutant line had a 4-bp insertion in the first exon (Fig. S4C). In addition, qRT-PCR results showed that the mRNA level of *TRAF4* decreased significantly (Fig. S4D). We first evaluated the survival rate of TRAF4$_4$$^{+/-}$ mosquitoes after fungal infection. Compared with the wild-type, the survival rate of TRAF4$_4$$^{+/-}$ mosquitoes significantly decreased (Fig. 4B), suggesting that these mosquitoes were less resistant to *B. bassiana*. Then, we studied the effect of TRAF4 on the translocation of Rel1 into nucleus. Immunostaining assays showed that after fungal infection, compared with wild-type mosquitoes, the Rel1 signal in nuclei was very weak in TRAF4$^{+/-}$ mosquitoes (Fig. 4C), suggesting that TRAF4 promotes Rel1 translocation into nucleus. Moreover, we studied the effect of TRAF4 on the expression of

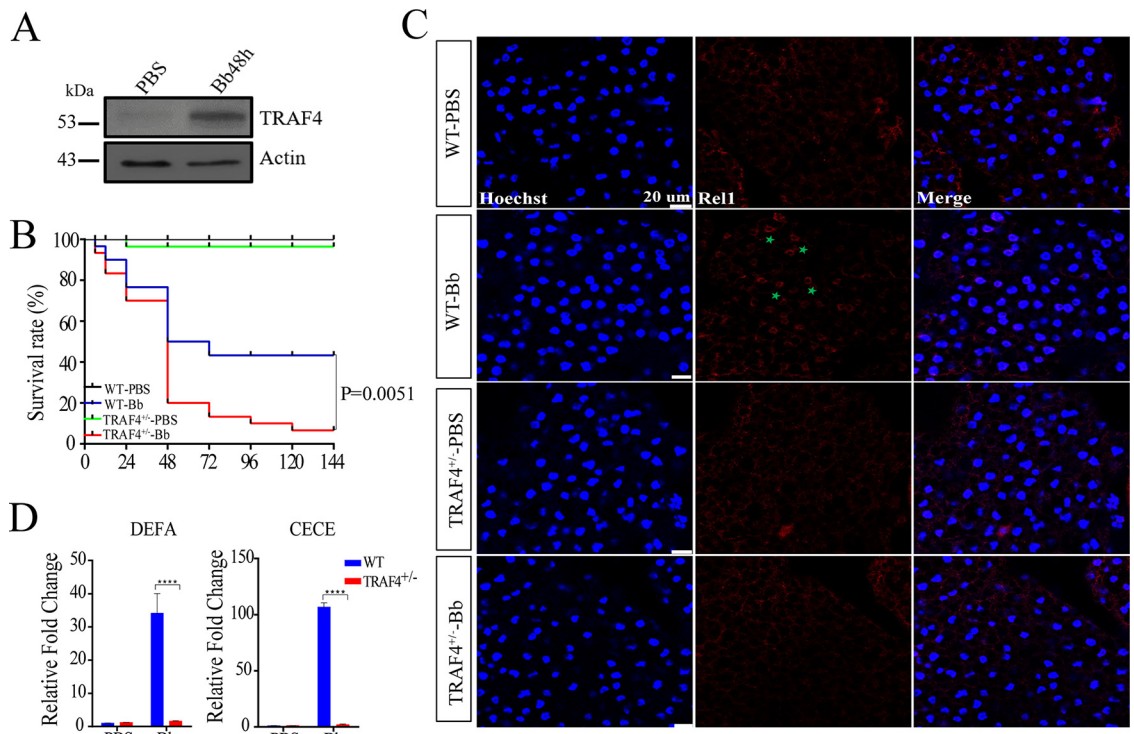

**FIG 4** Functional deletion of TRAF4 inhibits antifungal immunity of mosquitoes. (A) Immunoblotting detection of the expression of TRAF4. TRAF4 expression levels were detected using anti-TRAF4 antibody. Anti-actin antibody was used as control. (B) Survival assays showed the function of TRAF4 in mosquitoes after fungal infection. The TRAF4 heterozygote (TRAF$_4^{+/-}$) mosquitoes were more sensitive than wild-type (WT) to fungi ($P$ = 0.0163). Each experiment was performed in triplicate. (C) Immunofluorescence staining showed the localization of Rel1 (red) in the fat bodies treated with PBS or fungi (Bb). Scale bar, 20 $\mu$m. Green stars indicate Rel1 entering the nuclei. WT-PBS or WT-Bb, wild-type mosquitoes treated with PBS or fungi (Bb); TRAF$_4^{+/-}$-PBS or TRAF4$^{+/-}$ Bb, TRAF4 heterozygote mosquitoes treated with PBS or fungi (Bb). (D) qRT-PCR revealed the mRNA levels of *DEFA* and *CECE* after fungal infection. ****, $P < 0.001$; statistical analysis was performed using the Student's $t$ test. The experiments were repeated three times.

AMPs. qRT-PCR results showed that the mRNA levels of two selected AMPs were significantly reduced in TRAF4$^{+/-}$ mosquitoes after fungal infection, which was similar with that of TRAF4 RNAi mosquitoes, suggesting that TRAF4 activates the expression of AMP genes (Fig. 4D). Thus, our results confirm that TRAF4 functions upstream of NF-$\kappa$B in regulating mosquito immune responses.

**TRAF4 undergoes K63-linked polyubiquitination upon immune stimulation.** Several studies have confirmed that ubiquitin signaling plays important roles in the NF-$\kappa$B pathway (13). For example, the K63-linked polyubiquitin chains attached to TRAF6 could help the recruitment of TAK1 to mediate the activation of downstream pathways (21). In *D. melanogaster*, upon stimulation by peptidoglycan (PGN), Imd was K63-linked polyubiquitinated first, and then the ubiquitin chains were removed and the K48-polyubiquitin chains were linked, which mediated the degradation of Imd and the transmission of immune signals (22). To explore how TRAF4 participates in an antifungal immune response in mosquitoes, we examined whether TRAF4 is ubiquitinated upon immune challenge. Aag2 cells were transfected with pAc5.1b-TRAF4 plasmids and pAc5.1b-ubiquitin plasmids and then infected with Curdlan. Immunoblotting analysis showed that 6 h after treatment the expression of TRAF4 increased (Fig. S5A) and a higher level of ubiquitination was detected (Fig. S5B). At the same time, the mRNA levels of *DEFA* and *CECE* were significantly induced (Fig. S5C). Then, we investigated the types of polyubiquitin chains involved in TRAF4 ubiquitination. Plasmids with wild-type HA-tagged ubiquitin (WT) or mutants for different Lys residues (K6, K11, K27, K29, K33, K48, or K63) were cotransfected with V5-tagged TRAF4 into Aag2 cells. After incubating with Curdlan for 6 h, cell lysates were immunoprecipitated with anti-V5 antibody and analyzed by immunoblotting with anti-HA antibody. The sample transfected

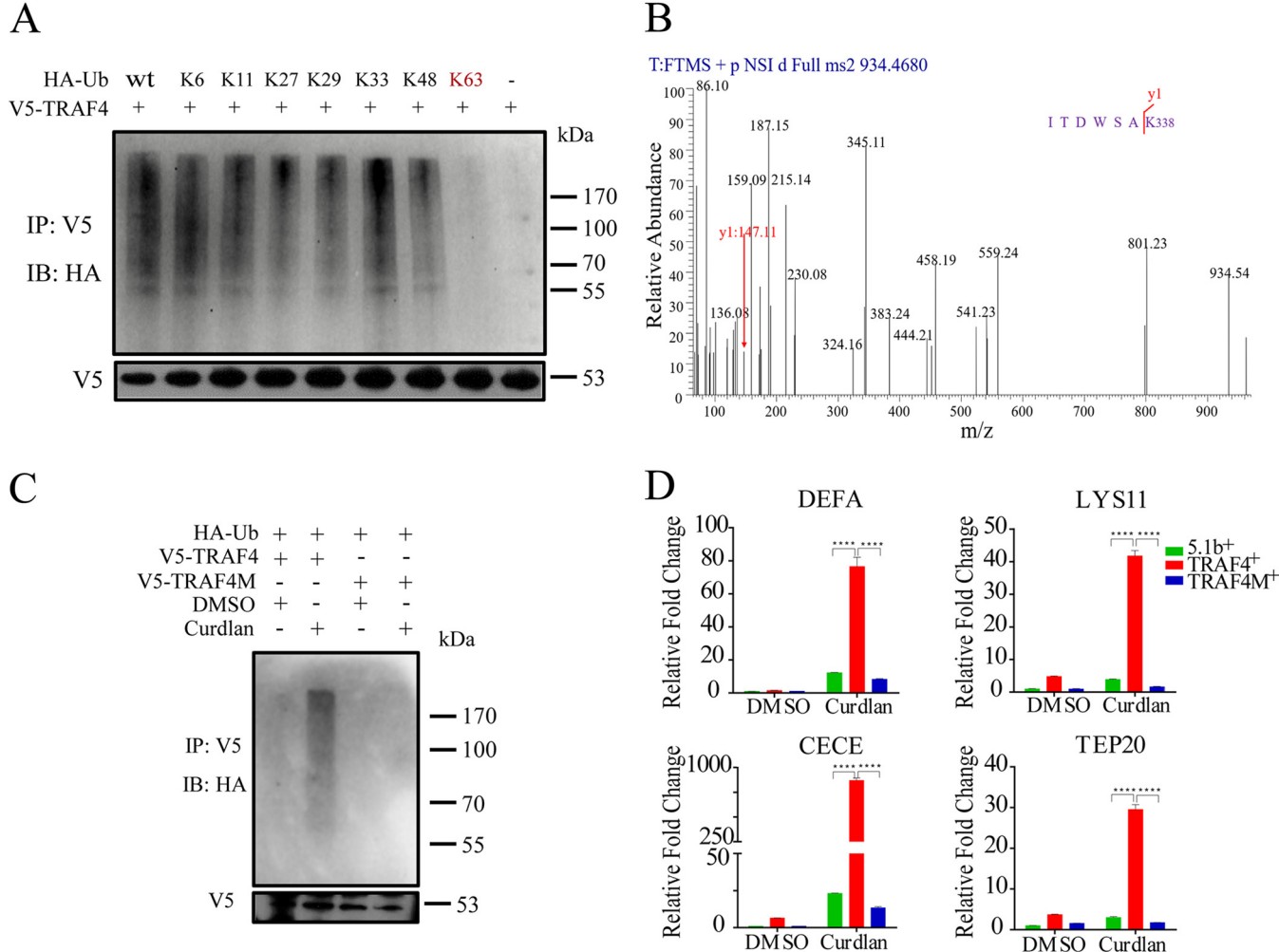

**FIG 5** K338 residue is the site for K63-linked ubiquitination of TRAF4. (A) Immunoblotting assays were performed to identify the link type of ubiquitin chains in TRAF4. Aag2 cells transfected with different plasmids were prepared to perform immunoprecipitation with anti-V5 antibody and analyzed by immunoblotting (IB) with anti-HA antibody. Anti-V5 antibody was used as control. (B) Identification the ubiquitination site of TRAF4. Overexpressed TRAF4 was immunopurified and analyzed using LC-MS/MS. Ubiquitination site (K338) was identified by their diglycine tags. Two biologically independent mass spectrometric analyses were conducted. (C) Inhibition of polyubiquitination of TRAF4 by mutation of K338 in Aag2 cells. IB shows the ubiquitination level of different treatment samples. Anti-V5 antibody was used as loading control. V5-TRAF4, V5-tagged TRAF4 plasmids; V5-TRAF4M, V5-tagged TRAF4 mutant (K338R) plasmids. (D) qRT-PCR results revealed the function of TRAF4 ubiquitination on the mRNA levels of selected immune effectors affected by the following: 5.1b$^+$, Aag2 cells overexpressiong pAC5.1b plasmids; TRAF4$^+$, Aag2 cells overexpressing TRAF4 plasmids; or TRAF4M$^+$, Aag2 cells overexpressing TRAF4 mutant plasmids. Data were normalized to the expression level of cells transfected with 5.1b$^+$ and treated with DMSO. ****, $P < 0.0001$; the Student's $t$ test was used for statistical analysis.

with only V5-tagged TRAF4 was used as a control. The results showed that a clear band was detected in each sample except for the K63 mutant (Fig. 5B), indicating that mutation of K63 affects the formation of polyubiquitin chains in TRAF4.

Next, the ubiquitin conjugation sites on TRAF4 were identified using immunoprecipitation and LC-MS/MS. The results showed that Lys residue 338 was conjugated with ubiquitin, as indicated by diglycine tags after trypsin digestion and MS/MS (y1, K338; Fig. 5B). To verify the ubiquitination site identified from mass spectrometry analysis, V5-tagged TRAF4 plasmids (V5-TRAF4) or mutant (338 Lys to Arg, K338R) TRAF4 plasmids (V5-TRAF4M) were cotransfected with HA-Ub plasmids into Aag2 cells. No ubiquitination band was detected after cell transfection with a mutant V5-TRAF4M (Fig. 5C). This suggested that K338 is the ubiquitination site of TRAF4. Moreover, we examined the function of TRAF4 ubiquitination on the expression of *DEFA*, *CECE*, *TEP20*, and *LYS11*. After treatment with Curdlan, the mRNA levels of these effector genes significantly increased in cells transfected with V5-TRAF4. However, the mRNA abundance of these genes after transfection with V5-TRAF4M was the same as that of the control group

(Fig. 5D). Taken together, these results indicate that K63-linked ubiquitination of TRAF4 is required for mosquito antifungal immunity.

**OTU7B modulates immunity by removing ubiquitin chains from TRAF4.** The above results have confirmed that K63-linked polyubiquitination of TRAF4 is essential to Toll pathway activation after fungal infection. To explore the possible involvement of OTU7B in the removal of K63-linked polyubiquitin chains from TRAF4, we first analyzed differences in total and K63 ubiquitination between wild-type and OTU7B$^{-/-}$ mosquitoes. No significant differences in total ubiquitination levels between wild-type and OTU7B$^{-/-}$ mosquitoes were observed after fungal infection (Fig. S6A), although the level of K63 ubiquitination was significantly higher in OTU7B$^{-/-}$ mosquitoes (Fig. S6B). We then immunoprecipitated TRAF4 from OTU7B$^{-/-}$ and wild-type mosquitoes infected with *B. bassiana*. The immunoblotting analysis revealed that the K63-linked polyubiquitin chains were obviously accumulated in OTU7B$^{-/-}$ mosquitoes (Fig. 6A) after fungal infection, suggesting that OTU7B is responsible for the removal of K63-linked polyubiquitin chains from TRAF4.

To further confirm whether OTU7B could remove ubiquitin chains from TRAF4, we measured its deubiquitinating activity. We expressed and purified the recombinant OTU7B (rOTU7B) protein in *Escherichia coli* (Fig. S6C). A fluorescence polarization assay was used to determine the release of a C terminal, isopeptide-linked fluorescent peptide (32). Ub-TAMRA was employed as a substrate in detecting OTU7B cleavage activity, as described previously (33, 34). Following normalization to the positive and negative-control samples, substrate-remaining curves for each concentration of rOTU7B were constructed (Fig. 6B). Purified rOTU7B showed the protein concentration-dependent DUB activity. *In vitro* assays proved that rOTU7B catalyzed the deubiquitinated reaction (Fig. 6C). All these results suggest that OTU7B catalyzes the reaction of removing ubiquitin chains from TRAF4.

We also examined the relationship between OTU7B and TRAF4 in the modulation of mosquito immune response after fungal infection. Immunostaining assays revealed that in OTU7B$^{-/-}$ mosquitoes after knockdown of TRAF4, the Rel1 signal in nuclei significantly decreased compared with control (Fig. 6D). These results were consistent with immunofluorescence in OTU7B mutant mosquitoes, as after TRAF4 knockdown, the nuclear Rel1 level declined following immune challenge (Fig. 6E). After treatment with EGFP dsRNA, the survival rate of OTU7B$^{-/-}$ mosquitoes showed only an incremental change relative to wild-type. However, in OTU7B$^{-/-}$ mosquitoes, when TRAF4 was knocked down, the survival rate sharply plummeted (Fig. 6F). Taken together, these findings show that OTU7B deubiquitinates TRAF4 thereby restricting its capacity to activate the Toll pathway.

## DISCUSSION

In this study, we investigated the role of ubiquitin system in the interaction between the entomopathogenic fungus *B. bassiana* and *Ae. aegypti* mosquitoes. *B. bassiana* infection dramatically induced the expression of DUB OTU7B. RNA-seq analysis revealed that expression of OTU7B resulted in the downregulation of the Toll pathway effectors. CRISPR-Cas9 knocking out of OTU7B increased mosquito survival after fungal infection. This suggests that OTU7B is involved in the counter effect against mosquito antifungal immunity. Therefore, we investigated the mechanism by which OTU7B restricts the Toll pathway. We identified the Toll adapter TRAF4 as a counterpart of OTU7B. Our results showed that, after fungal infection, OTU7B interacts with TRAF4 and causes the removal of its ubiquitin chains.

Mammals and insects have developed an array of mechanisms that enable them to generate a rapid defensive response to protect themselves from the attack of various pathogens. Mammalian Toll-like receptors and *D. melanogaster* Toll mediate the conserved signal transduction pathway, which plays an essential role in innate immune defense in vertebrate and invertebrate organisms (35, 36). In turn, some pathogens have evolved effective strategies to suppress immune responses. For instance,

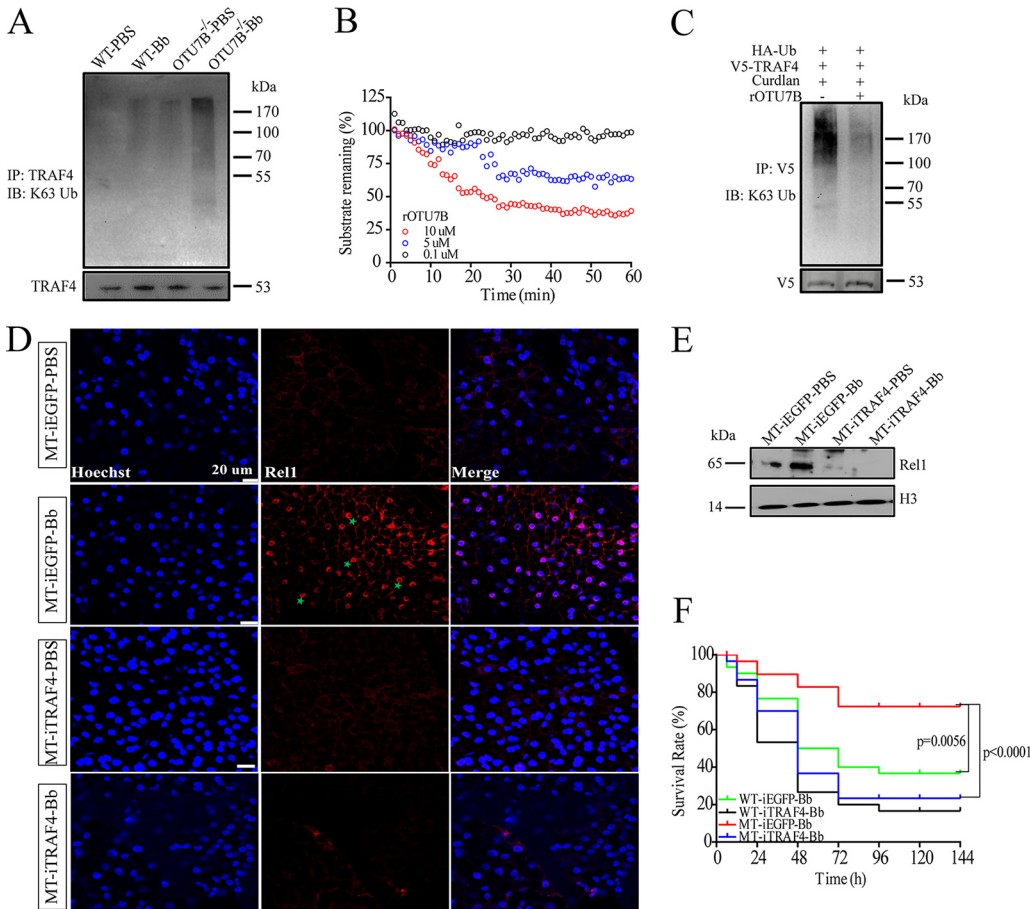

**FIG 6** OTU7B removes K63-linked polyubiquitin chains from TRAF4. (A) The ubiquitination levels of TRAF4 in mosquitoes. Proteins extracted from differently treated mosquitoes were used to perform immunoprecipitation with anti-TRAF4 antibody and analyzed by immunoblotting (IB) with anti-K63 Ub antibody. Anti-TRAF4 antibody was used as loading control. WT-PBS or WT-Bb, wild-type mosquitoes treated with PBS or fungi (Bb); OTU7B$^{-/-}$-PBS or OTU7B$^{-/-}$-Bb, OTU7B mutant mosquitoes treated with PBS or fungi (Bb). (B) The deubiquitination activity of OTU7B was detected by fluorescence polarization assays of Ub-TAMRA cleavage. The deubiquitination activity of different concentrations of recombinant OTU7B (rOTU7B) was measured. (C) The removal of ubiquitin chains from TRAF4 by OTU7B was detected *in vitro*. Lysates of Aag2 cells overexpressing V5-TRAF4 plasmids were used to perform immunoprecipitation with anti-V5 antibody, and then rOTU7B was added before IB analysis with anti-K63 Ub antibody. Anti-V5 antibody was acted as control. (D) Immunofluorescence staining showed the effects of TRAF4 on the localization of Rel1 (red) in the fat bodies of OTU7B mutant-type mosquitoes (MT) after fungal infection (Bb). Scale bar, 20 $\mu$m. (E) Immunoblotting results show the expression of Rel1 in the nucleus of mosquitoes subjected to different treatments. Anti-histone3 (H3) antibody acted as control. (F) Survival assays show that OTU7B inhibited the antifungal immunity of mosquitoes through TRAF4. After EGFP dsRNA treatment, the survival rate of MT was higher than that of wild-type mosquitoes (WT) during *B. bassiana* infection. When TRAF4 was knocked down (MT-iTRAF4), MT mosquitoes were more sensitive to *B. bassiana* than the controls (MT-iEGFP) ($P < 0.0001$). Each experiment was performed in triplicate. MT-iEGFP or MT-iTRAF4, OTU7B$^{-/-}$ mosquitoes treated with EGFP or TRAF4 dsRNA; WT-iEGFP or WT-iTRAF4, wild-type mosquitoes treated with EGFP or TRAF4 dsRNA.

pathogens can utilize the host immune system by regulating the ubiquitin system, which is a critical regulator of pathogen-host interactions (14). Many bacteria effectors have been found to act as E3 ligases or DUBs to modulate host ubiquitination signaling (37). To evade host defense, *Shigella flexneri* bacteria use the effector IpaH9.8, an E3 ligase, promoting deubiquitination and degradation of interferon-inducible guanylate-binding proteins (38).

DUBs are a family of proteins that counterregulate ubiquitination. The ubiquitin-editing enzyme A20 and the DUB CYLD are widely recognized enzymes that are involved in the inhibition of innate immune responses (39). In mammals, the DUB MYSM1 negatively regulates the immune reaction by removing the ubiquitin chains of the TRAF3 and TRAF6 complexes (40). Similar to A20, OTU7B contains an OTU domain at its N terminal and is considered to have DUB enzymatic activity. Our study has

revealed that OTU7B was induced after fungal challenge (Fig. 1A) and prevented Rel1 from nuclear entry (Fig. 2B), suggesting an important role in regulation of mosquito antifungal immunity.

Recent investigations revealed new roles of ubiquitination in protein trafficking and regulation of kinases and phosphatases (41), making it a critical regulator of immune signaling, particularly in conserved NF-$\kappa$B pathways (12). In insects, studies on how ubiquitination regulates immunity mainly focus on the IMD pathway. For example, ubiquitination of Imd in *D. melanogaster* mediates antibacterial immunity (22). The *D. melanogaster* ubiquitin-specific protease 36 (USP36) and USP2 negatively regulate the IMD pathway after infection by Gram-negative pathogens (42, 43). Compared with the IMD pathway, there are a few studies on ubiquitination regulating the Toll pathway. For example, the loss of function of the *Drosophila* ubiquitin-conjugating enzyme 9 (dUbc9) results in the expression of *drosomycin* and *cecropin*, although its underlying mechanism remains unknown (44). A previous study revealed that Pellion, a RING-containing ligase, inhibits the Toll pathway by mediating K48-linked ubiquitination and degradation of MyD88 (45). Whether ubiquitination is involved in regulating other components of the Toll cascade remains to be investigated. The Toll pathway has two phases based on the condition of TRAF4. It is turned on when TRAF4 is K63 ubiquitinated. In this way, the sense of any microbes by peptidoglycan recognition protein (PGRP) and $\beta$-1,3-glucan recognition protein ($\beta$GRP) could induce the synthesis of AMP via Toll. We have shown here that the pathogenic *B. bassiana* is able to manipulate the mosquito immune defense system by activating the DUB OTU7B, which in turn causes TRAF4 deubiquitination. As a result, the Toll pathway is switched off, and the mosquito becomes more sensitive to *B. bassiana* invasion.

TRAFs constitute a family of conserved signal adaptor proteins that have been found in mammals (46) and other multicellular organisms, such as *D. melanogaster* (31, 47). In general, TRAF proteins possess E3 ubiquitin ligase and scaffolding functions, both of which are involved in signal transduction (48, 49). In mammals, most TRAF proteins contain a RING finger at the N terminal, which is important for E3 ligase function, while the scaffolding function is mainly based on the TRAF domain, which mediates the interactions between membrane receptors and a variety of downstream molecules (48). In *Ae. aegypti*, 22 proteins with TRAF domains were identified, 8 of which contain RING. In our study, we identified that the OTU7B-interacting TRAF4 has a RING domain at its N terminus, suggesting that it might act via a dual function. We have determined that TRAF4 was ubiquitinated at K338 to form K63-linked polyubiquitin chains, which is important for the induction of immune effectors after fungal infection (Fig. 5A to D). Besides, the DUB OTU7B can remove K63-linked polyubiquitin chains from TRAF4 (Fig. 6A to C), while knockdown of TRAF4 in OTU7B$^{-/-}$ mosquitoes increased sensitivity to fungi (Fig. 6D and F). These results indicate that OTU7B/TRAF4 is a novel inducible regulatory switch of the Toll pathway. We also found that the homozygous mutants of TRAF4 are lethal, indicating that TRAF4 may play other essential roles in mosquitoes.

In conclusion, we identified a mechanism by which the fungus *B. bassiana* affects the mosquito innate immunity (Fig. 7). We have shown that the fungal infection triggers the expression of DUB OTU7B, which enzymatically removes K63-linked polyubiquitin chains from the Toll pathway adapter TRAF4, preventing the translocation of the NF-$\kappa$B factor Rel1 into the nucleus, and mosquitoes become more sensitive to fungi. Further exploration of this uncovered mechanism should lead to the improvement of the *B. bassiana* effectiveness as a biopesticide.

## MATERIALS AND METHODS

**Animals and cells.** The Liverpool strain of *Ae. aegypti* mosquito was raised in our laboratory in a biological incubator according to the precious methods (50). Adults were fed with water and 10% sucrose solution. For the production of eggs, mosquitoes were fed with chicken blood. All procedures using vertebrate animals were approved by the Animal Care and Use Committee of Institute of Zoology, Chinese Academy of Sciences (Beijing, China). *Ae. aegypti* Aag2 cells cultured at 27°C in Schneider's *Drosophila* medium (Invitrogen) supplemented with 8% heat-inactivated fetal bovine serum were used for

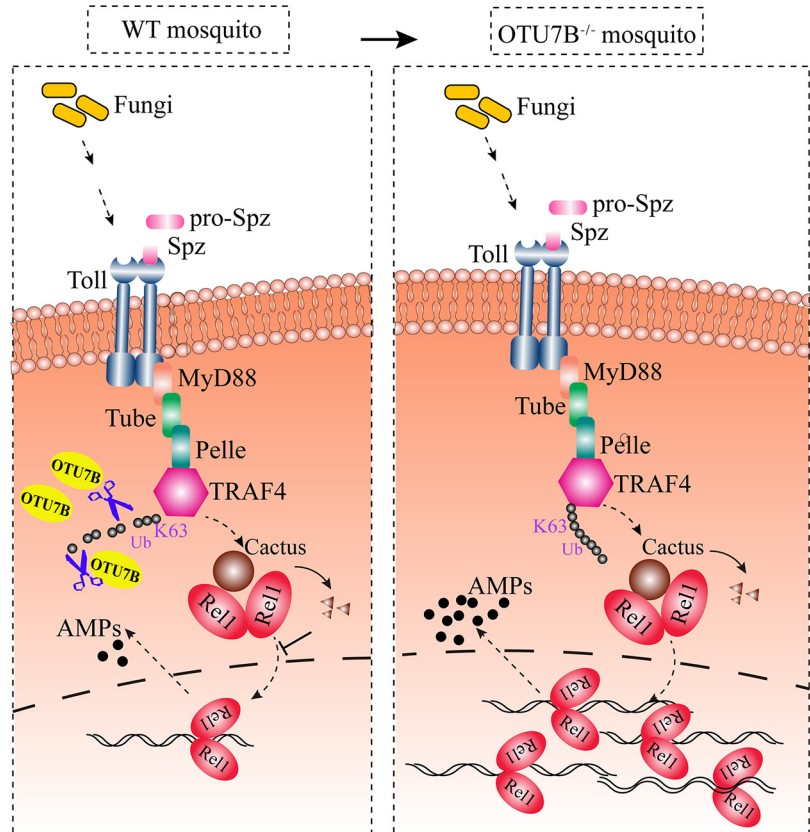

**FIG 7** Schematic diagram of the mechanisms of OTU7B-mediated antifungal immunity in *Ae. aegypti*. In wild-type (WT) mosquitoes, after fungal infection, the expression of OTU7B was upregulated, and then OTU7B cleaved the K63-linked ubiquitin chains from TRAF4, preventing the translocation of Rel1 into the nucleus, blocking the expression of AMPs. In OTU7B mutant (OTU7B$^{-/-}$) mosquitoes, after fungal infection, Rel1 translocated into the nucleus, leading to a lot of AMP expression, which can kill fungus.

transfection and immune challenge. Septic injury by *B. bassiana* ARSEF2860 ($5 \times 10^7$ conidia/mL) was performed as described elsewhere (51).

**Synthesis and microinjection of dsRNA.** RT-PCR was used to obtain the cDNA templates (400-bp to 1-kb in length) of target genes. dsRNA synthesis was performed using T7 RiboMAX Express RNAi kit (Promega) according to the manufacturer's protocol. The EGFP gene was used as control. Approximately 1 $\mu$g dsRNA was transferred into the thorax of ice-anesthetized mosquitoes by an injector (Nanoliter 2000; World Precision Instrument). RNAi efficiency was confirmed using qRT-PCR. All primers used are shown in Table S5.

**Survival rate analysis.** The newly emerged females were injected with different dsRNA based on various experimental designs. Three days later, mosquitoes (30/group) were challenged with *B. bassiana* or PBS. The survival curves were compared using GraphPad 6.0 software. $P$ value thresholds were calculated using log-rank or Mantel-Cox tests, and differences with $P < 0.05$ were regarded as statistically significant.

**Dual-luciferase reporter assay.** The promoter sequence of *OTU7B* ($-1,940$ to $+186$) was cloned into luciferase reporter vector pGL4.10 (pGL4.10-OTU7B). The open reading frame of *Rel1A/Rel1B* was inserted into the pAc5.1b plasmids to construct overexpression vectors, respectively (pAc5.1b-Rel1A/Rel1B). Then, pGL4.10-OTU7B and pAc5.1b-Rel1A/Rel1B plasmids were cotransfected with pGL4.73 vector into Aag2 cells using FuGENE 6 (Promega). Forty-eight hours later, cells were collected to determine the luciferase activity using the Dual-Luciferase Reporter Assay System (Promega) according to the manufacturer's procedure.

**EMSA.** pAc5.1b and pAc5.1b-Rel1A plasmids were transfected into Aag2 cells using FuGENE 6 (Promega). After 48 h, cells were collected to extract nuclear protein for EMSA following the instruction of the Pierce Light Shift Chemiluminescent EMSA kit (Thermo Fisher Scientific). Oligonucleotide probes used are provided in Table S5.

**RNA-seq analysis and qRT-PCR.** The high-throughput sequencing platform HiSeq 2000 was used to sequence libraries that were constructed from treated fat bodies using an Illumina kit v2. For the following bioinformatics analysis, a flux-capacitor was used to determine the reads count (FPKM) of the transcripts, and the DEGseq package in R language was employed to identify DEGs (52). The reference genome used in this experiment was downloaded from the VectorBase website (https://vectorbase.org/

vectorbase/app/downloads/release-47/AaegyptiLVP_AGWG/fasta/data/). Genes were considered differentially expressed when the $P$ value was <0.05. Cross comparison of each treated sample was normalized by its FPKM. The iEGFP sample was treated as the background when the fold change rate was calculated. The transcriptome data were deposited to the NCBI SRA (bioproject PRJNA744689). qRT-PCR was performed using SuperReal PreMix (Tiangen) on StepOnePlus (Thermo Fisher Scientific). Template concentrations were normalized to the internal reference $RPS7$ gene. The primers used in this experiment are listed in Table S5.

**Embryonic injection and mutant line generation.** sgRNAs were designed using a web-based tool (https://zlab.bio/guide-design-resources) and synthesized using the T7 RiboMAX Express RNAi kit (Promega). Then, sgRNAs were purified using the MEGAclear Transcription Clean-Up kit (Ambion), according to the manufacturer's protocol. Next, sgRNAs (50 ng/$\mu$L) and Cas9 protein (333 ng/$\mu$L) (PNA Bio) were incubated for 20 min at 37°C and then microinjected into the posterior pole of embryos using Eppendorf FemtoJet 4i. The embryos were hatched at 5 days after injection and reared according to described protocols. All females of the $G_0$ generation were identified by sequencing. Mosaic female mosquitoes were crossed with wild-type (WT) males and blood fed for oviposition. The $G_1$ (offspring of mosaic females and WT males) were hatched and reared to adulthood. Heterozygotes were identified by sequencing and crossed with WT. At the same time, the heritability of the genotype was confirmed by cloning and sequencing. Heterozygotes with the same genotype were selected from $G_2$ (offspring of heterozygote and WT). Homozygotes were obtained after heterozygous selfing at $G_3$. The knockout line was obtained after at least three generations of homozygous selfing.

**Immunoprecipitation and immunoblotting analysis.** For immunoprecipitation, Aag2 cells were transfected with indicated plasmids and suspended in lysis buffer containing 50 mM Tris-HCl (pH 7.5), 1% Nonidet P40, 0.5% sodium deoxycholate, 150 mM sodium chloride, 1 tablet complete protein inhibitor cocktail/50 mL, and 0.7 $\mu$g/mL pepstatin. To reduce the nonspecific binding of unrelated proteins to protein A-agarose (Roche), a 50-$\mu$L suspension of homogeneous agarose beads was mixed with the sample, incubated at least 3 h at 2°C to 8°C, and then centrifuged for 20 s at 12,000 $\times$ $g$. The supernatant was separated and incubated with specific antibodies at 4°C overnight. The beads with complex were pulled down and washed with lysis buffer, washing buffer 2, and washing buffer 3, according to the producer's instructions. After centrifugation, the agarose beads were resuspended in 2$\times$ SDS loading buffer, and proteins were heat denatured. After removal of the agarose, the mixture was subjected to immunoblotting with different antibodies (all antibodies used are listed in Table S6). Ubiquitination assays were performed as described elsewhere (53). After homogenizing in lysis buffer, the tissue samples were centrifuged. The supernatant was mixed with SDS sample buffer and then heated before immunoblotting with anti-ubiquitin antibodies.

**Immunofluorescence microscopy.** Fat bodies from differently treated mosquitoes were fixed with 4% paraformaldehyde and then permeabilized with PBS containing 0.5% Triton X-100 for 10 min. Tissues were gently washed with PBS three times, blocked for 1 h at room temperature (RT) with 3% BSA, and incubated with indicated antibodies (1:700) overnight at 4°C. The tissues were washed with PBS again and then incubated with fluorescence-labeled antibodies (Alexa Fluor 488, green, A11034, goat-anti-rabbit; 546, red, A21123, goat-anti-mouse; 594, red, A11037, goat-anti-rabbit) (1:1,000) for 2 h at RT, washed with PBS, and stained with Hoechst (H3570; Life Technologies) at RT for 10 min, washed with PBS, mounted onto microscope slides, and finally imaged using Zeiss LSM710.

**GST pulldown.** The pGEX-4T-Pelle (GST-Pelle) fusion protein was used as a bait protein. The pET28a-TRAF4 (His-TRAF4) fusion protein acted as a prey protein, and pET28a and pGEX-4T vectors were regarded as controls. The samples were washed with PBS six times; 1 mL bacteria supernatant containing GST or GST-Pelle was added into glutathione Sepharose 4B beads (GE Healthcare). After incubating for 2 h at 4°C, the supernatant was separated. The beads were washed six times with PBS with 1% Triton X-100. Bacterial lysate of His-TRAF4 or His-pET28a was added to the beads for 4 h at 4°C and then washed as above to remove the unbound protein. The protein bind on the beads was examined using immunoblotting. Anti-His (1:5,000) and anti-GST (1:3,000) antibodies were used.

**Identification of ubiquitination sites.** Aag2 cells were transfected with pAC5.1b-V5-TRAF4 (V5-TRAF4) for 24 h, then incubated with Curdlan (0.2 mg/mL) and collected after 6 h. The cell lysates were immunoprecipitated with anti-V5 antibody. After being separated by SDS-PAGE, the corresponding bands were analyzed by LC-MS/MS (Beijing Protein Innovation, Beijing); two technical replicates were performed.

To determine the type of ubiquitination, wild-type HA-ubiquitin (HA-Ub) or mutants at different Lys residues (K6, K11, K27, K29, K33, K48, or K63) were cotransfected into Aag2 cells with V5-TRAF4 plasmids and treated with Curdlan. The cell lysates were immunoprecipitated using anti-V5 antibody, followed by immunoblotting using anti-HA antibody.

**Determination of OTU7B cleavage activity.** Fluorescent Ub-TAMRA substrates were obtained from LifeSensors. Cleavage activity of recombinant OTU7B (rOTU7B) was measured by means of fluorescence polarization (FP), as previously described (33, 34). A dilution series was prepared using purified rOTU7B in dilution buffer (25 mM Tris-HCl [pH 7.4], 5 mM $\beta$-mercaptoethanol, 100 mM NaCl, and 0.1 mg/mL BSA) at twice the final concentration and incubated at RT for 15 min. rOTU7B was prepared at final concentrations of 0.1, 5, and 10 $\mu$M. Fluorescent Ub-TAMRA substrate was prepared at 200 nM. Then, 10 $\mu$L of rOTU7B and substrates were mixed in a 384-well black plate (PerkinElmer; LBS-coated OptiPlate) in triplicate. Control samples included dilution buffer only (black), Ub-TAMRA only (negative control), and TAMRA (50 nM) only (positive control). FP values were measured at RT on a reader (PerkinElmer, Envision) at wavelengths of 531 nm and 595 nm, and normalized to the positive and negative controls to construct the percentage-substrate-remaining curve (34).

***In vitro* analysis of deubiquitination.** Aag2 cells transfected with V5-TRAF4 and HA-Ub were incubated with Curdlan for 6 h. The cell lysates were incubated with anti-V5 antibody (the procedure of immunoprecipitation is as described above). For deubiquitination assays, the beads were resuspended in deubiquitination buffer containing 25 mM $MgCl_2$, 125 mM HEPES (pH 7.5), 10 mM NaF, 2.5 mM DTT, 10 nM okadaic acid, 500 mM NaCl, and 0.5 mg/mL BSA. Then, 10 $\mu$L rOTU7B (5 $\mu$g) was added to the beads and incubated for 30 min with gentle shaking. The mixture was analyzed by immunoblotting with anti-K63 Ub antibody.

**Statistical analysis.** All data are displayed as the mean $\pm$ SEM. For qRT-PCR, the Student's *t* test was used for statistical analysis. For survival analysis, the survival curves were generated using Kaplan-Meier methods, and the *P* value was calculated using the log-rank or Mantel-Cox test. GraphPad 6.0 software was used for all statistical analyses.

**Data availability.** All data required to evaluate the conclusions of the paper are available in the paper and/or the Supplemental Material. RNA-seq data generated in this paper have been uploaded to the NCBI SRA (bioproject PRJNA744689). The mass spectrometry proteomics data have been deposited to the ProteomeXchange Consortium (http://proteomecentral.proteomexchange.org) with the data set identifier PXD038261. Other data related to this article may be obtained from the authors.

## SUPPLEMENTAL MATERIAL

Supplemental material is available online only.

**SUPPLEMENTAL FILE 1**, PDF file, 1.7 MB.

## ACKNOWLEDGMENTS

This work was supported by grants from the National Science Foundation of China (32090011), National Key Plan for Scientific Research and Development of China (2021YFC2600100), and Strategic Priority Research Program of Chinese Academy of Sciences (XDPB16).

We declare that we have no competing interests.

We thank Haobo Jiang from Oklahoma State University for valuable suggestions about the manuscript.

Y. Wang, A. S. Raikhel, and Z. Zou designed research; Y. Wang, M. Chang, M. Wang, and Y. Ji performed research; Y. Wang, A. S. Raikhel, and Z. Zou wrote the manuscript.

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
