## [Reviewer comments · Microbiology Spectrum]

Microbiology Spectrum

OTU7B modulates the mosquito immune response to *Beauveria bassiana* infection via deubiquitination of the Toll adaptor TRAF4

Yan-Hong Wang, Meng-Meng Chang, Mao Wang, Yan-Nan Ji, Xiao-Mei Sun, Alexander Raikhel, and Zhen Zou

Corresponding Author(s): Zhen Zou, Institute of Zoology, Chinese Academy of Sciences

Review Timeline:

Submission Date:	August 19, 2022
Editorial Decision:	October 1, 2022
Revision Received:	November 23, 2022
Accepted:	November 29, 2022

Editor: Chengshu Wang

Reviewer(s): Disclosure of reviewer identity is with reference to reviewer comments included in decision letter(s). The following individuals involved in review of your submission have agreed to reveal their identity: Zongzhao Zhai (Reviewer #1)

Transaction Report:

DOI: <https://doi.org/10.1128/spectrum.03123-22>

October 1, 2022

Prof. Zhen Zou
Institute of Zoology, Chinese Academy of Sciences
Beijing
China

Re: Spectrum03123-22 (OTU7B modulates the mosquito immune response to Beauveria bassiana infection via deubiquitination of the Toll adaptor TRAF4)

Dear Prof. Zhen Zou:

Link Not Available

Sincerely,

Chengshu Wang

Journals Department
Reviewer comments:

Reviewer #1 (Public repository details (Required)):

RNA-seq
protein identification by MS

Reviewer #1 (Comments for the Author):

This ms from Zou and colleagues identified a novel role of TRAF4 in mosquito Toll pathway activation, and demonstrated that the function of TRAF4 in Toll is antagonized by its binding partner OTU7B which removes the K63-ub from TRAF4. Using mosquito cell culture, they further show that K63-linked ubiquitination of TRAF4 is required for Toll signaling. As the

(de)ubiquitination system has not been previously studied for Toll-dependent insect immune response, the identification of TRAF4 as positive regulator and OTU7B as a negative regulator of the mosquito Toll signaling upon fungal infection is particularly exciting. Therefore, this paper should attract a broad interest beyond the mosquito community. The experiments are generally well performed. However, several issues need to be clarified before publication, as follows. The writing of the paper also needs to be significantly improved.

TRAF4 as an adaptor protein that activates the Toll pathway is not experimentally verified. To my knowledge, although TRAFs were suggested to interact with Pelle, insect TRAFs have never been experimentally associated with immunity. It should greatly enhance the novelty of current ms by better placing TRAF4 into the Toll signaling cascade, e.g. by providing data showing that TRAF4 physically interacts with Pelle during mosquito Toll activation upon microbial challenge.

As OTU7B and TRAF4 form a protein complex and show very similar cellular localization during immune challenge, it is interesting to test if their localization is inter-dependent using the CRISPR mutants this paper generated.

Regarding the interpretation of Relish staining, Rel1 signal was actually in the nuclear (determined by co-localization with Hoechst staining) but not peri-nuclear or around the nuclear as the authors interpreted. This concerns Fig 2A, 4C and 6D. It is intriguing that Rel1 staining was mainly accumulated in the peri-nuclear compartment of the fat body cells in OTU7B^{-/-} Bb group. Judging from the ratio of perinuclear to nuclear Relish, an alternative interpretation of this figure is that loss of OTU7B in fact blocks the nuclear translocation of Relish1, in contrast to what the authors have proposed in the paper. Moreover, the chromatin seems to be condensed in OTU7B^{-/-} Bb fat body cells, forming a quite different nuclear structure compared to the other three groups shown in the same figure. Are OTU7B^{-/-} Bb fat body cells undergoing apoptosis?

The effect of TRAF4 heterozygous mutation on the two selected AMP expression is unusually strong (Fig 4D). It implies that loss of only one copy of TRAF4 gene is already able to completely abolish Toll signaling activation. Any explanation on this? As CRISPR often generates secondary mutations, the authors should measure Toll activation via TRAF4-specific RNAi to further support their claim.

Regarding the writing:

- Curdian should be introduced for non-mosquito audience.
- The use of "the" needs to be thoroughly revised.
- Use either "...K63-linked polyubiquitination" or "xxx is K63-polyubiquitinated at...", rather than "K63-link polyubiquitinated"
- I suggest precise terms to be used when making conclusions. e.g. line 214, use restricting rather than affecting; line 314, change "plays an important role in " to "is required for".
- Line 73, "cov unterattacks" is counterattacks?
- Line 88, Ubiquitination, the ligation reaction catalyzed by ubiquitin ligase to removal by deubiquitinases (DUBs)..... Please re-write it.
- Line 93, "The process of ubiquitination requires the coordination of three kinds of enzymes: E1 activating, E2 conjugating, and E3 ligating." Need to re-write.
- Line 107, IMD does not stand for Immunodeficient, but rather it is abbreviation of immune deficiency.
- Line 122, OUT DUBs is OUT-domain containing DUBs?
- Line 242, "in contrast" should be "similarly".
- Line 175, change to ".....genes involved in lipid metabolism increased after OTU7B knockdown (Fig. S2D), consistent to the notion that immune response is an energy-consuming process (26)".
- line 297, K63 mutation affects the enzymatic formation of polyubiquitin chains, please re-write.
- line 347, "...OTU7B inhibits TRAF4 from functioning in the Toll pathway by its deubiquitination" to "...OTU7B deubiquitinates TRAF4 thereby restricting its capacity to activate the Toll pathway".
- Line 396, change to "PGRP and β GRP". What is β GRP?
- Line 412, K63-linked polyubiquitination chains, change to "...polyubiquitin chains"
- Line 420, "In conclusion, we identified the mechanism by which the fungus *B. bassiana* affects the mosquito innate immunity (Fig. 7)." I believe this is a (nice) mechanism, but not the (only) mechanism.

Reviewer #2 (Comments for the Author):

Mosquitoes use innate immunity to combat microorganisms, and reversely microorganisms are used to control mosquito population. The authors found that an entomopathogenic fungus induced deubiquitinase OTU7B, which, when knocked out, enhanced resistance to fungal infection via the NF- κ B factor, Rel1. They further revealed that OTU7B cleaves K63-linked ubiquitin chains from TRAF4, the integrative part of Toll pathway. Thus, mosquitoes become much more sensitive to the *Beauveria bassiana* infection. This study uncovered the novel mechanism of fungal action against the host innate immunity. It gives a way to improve the effective entomopathogen. This study is well conducted, and the data support the conclusions. Great efforts were made to prove this, including preparing two Crispr-cas9 knockout lines, RNAseq, and detailed biochemical analysis. In general, The work represents potentially interesting insights in mosquito immune response to fungus. I recommend the article

for publication with some minor modifications.

Major comments:

- 1, In Fig1 A-C, the authors proved that, after fungal infection, the expression of OTU7B increases significantly. However, it would be clearer, if the author could give a detailed mechanism of how OTU7B is induced to express.
- 2, In figure 2A, the author only showed the translocation of Rel1 into nucleus at 48 h after fungal infection, why? Does the author have any results of other infection time?

Minor comments:

- 1, The authors should check the English writing of the whole article carefully.
- 2, The author used "ubiquitin chains" in abstract and "polyubiquitin chains" in the whole manuscript. It is suggested to modify.
- 3, Line 103 and 107, "K63-linked" or "K63-link"? check it.
- 4, Line 115, change "ovarian tumor" to "ovarian tumor (OTU)".
- 5, Line 141, change "Bb infection" to "B. bassiana infection".
- 6, Line 182, I think Fig 1C should be 1F, and line 187, Fig 1D should be 1G.
- 7, In page 7, line 192, change "Ae. aegypti Rel1(AaRel1)" to "AaRel1".

Staff Comments:

Preparing Revision Guidelines

Please return the manuscript within 60 days; if you cannot complete the modification within this time period, please contact me. If you do not wish to modify the manuscript and prefer to submit it to another journal, please notify me of your decision immediately so that the manuscript may be formally withdrawn from consideration by Microbiology Spectrum.

November 10, 2022

Microbiology Spectrum

Dear Editor,

Thank you very much for your consideration and help. We truly appreciate yours and all reviewers' constructive comments. All comments are valuable and very helpful for improving this manuscript. We revised the manuscript according to these suggestions, incorporating additional experiments requested by reviewers. Below, please find detailed point-by-point answers.

Reviewer #1

1. Public repository details (Required): RNA-seq, protein identification by MS.

Thank you for your suggestion. We added the details of public repository used in RNA-seq and MS in the manuscript and listed the dataset number of RNA-seq and MS in the "Data availability" part.

The reference genome used in this experiment was downloaded from website (<https://vectorbase.org>).

RNA-seq data generated in this paper have been uploaded to the NCBI SRA (bioproject PRJNA744689).

The protein database used in MS identification was downloaded from website (https://vectorbase.org/vectorbase/app/downloads/release-47/AegyptiLVP_AGWG/fasta/data/).

The mass spectrometry proteomics data have been deposited to the ProteomeXchange Consortium (<http://proteomecentral.proteomexchange.org>) with the dataset identifier PXD038261.

2. TRAF4 as an adaptor protein that activates the Toll pathway is not experimentally verified. To my knowledge, although TRAFs were suggested to interact with Pelle, insect TRAFs have never been experimentally associated with immunity. It should greatly enhance the novelty of current ms by better placing TRAF4 into the Toll signaling cascade, e.g. by providing data showing that TRAF4 physically interacts with Pelle during mosquito Toll activation upon microbial challenge.

Many thanks to the reviewer's proposal. We performed a new experiment to study the interaction of Pelle and TRAF4. As shown in Fig. S3C, we confirmed that Pelle and TRAF4 can interact in vitro through GST pull-down assay. This helps us to further study the role of TRAF4 in the Toll pathway.

3. As OTU7B and TRAF4 form a protein complex and show very similar cellular localization during immune challenge, it is interesting to test if their localization is

inter-dependent using the CRISPR mutants this paper generated.

We added a new experiment to study the localization of TRAF4 in OTU7B mutant mosquitoes during fungal infection. As shown in Fig. S3D, in OTU7B^{-/-} mosquitoes, after fungal infection, the localization of TRAF4 is different from that of wild type mosquitoes, indicating that OTU7B restricts the distribution of TRAF4 in immune response.

4. Regarding the interpretation of Relish staining, Rel1 signal was actually in the nuclear (determined by co-localization with Hoechst staining) but not peri-nuclear or around the nuclear as the authors interpreted. This concerns Fig 2A, 4C and 6D. It is intriguing that Rel1 staining was mainly accumulated in the peri-nuclear compartment of the fat body cells in OTU7B^{-/-} Bb group. Judging from the ratio of perinuclear to nuclear Relish, an alternative interpretation of this figure is that loss of OTU7B in fact blocks the nuclear translocation of Relish1, in contrast to what the authors have proposed in the paper. Moreover, the chromatin seems to be condensed in OTU7B^{-/-} Bb fat body cells, forming a quite different nuclear structure compared to the other three groups shown in the same figure. Are OTU7B^{-/-} Bb fat body cells undergoing apoptosis?

Thank the reviewer for the valuable comments, which are very important to us. The previous figures (Fig2A, 4C and 6D) are easy to be misunderstood. We referred to Tanji's paper (Tanji. et al, PNAS, 2010), changed the experimental strategy (changed the magnification from 40x to 20x, which helps reduce the ambiguity), and redone the fluorescence experiments. The new figures can better confirm the proposal of our article. In addition, a new immunoblotting experiment was also added, as shown in Fig 6E, which further confirmed that the interaction between OTU7B and TRAF4 restricts the translocation of Rel1 during fungal infection.

We used the transcriptome data to analyze the expression profile of apoptosis related genes (Figure below). We found that, after reducing the expression of OTU7B, the mRNA level of apoptosis genes (such as Caspase 1 and 2) increased, while at the same time, the expression of apoptosis inhibitor genes (such as IAP1) also increased. At present, it is hardly to determine whether apoptosis occurred, and further experimental verification will be conducted later.

5. The effect of TRAF4 heterozygous mutation on the two selected AMP expression is unusually strong (Fig 4D). It implies that loss of only one copy of TRAF4 gene is

already able to completely abolish Toll signaling activation. Any explanation on this? As CRISPR often generates secondary mutations, the authors should measure Toll activation via TRAF4-specific RNAi to further support their claim.

Thanks for the reviewer's comments. In Fig 4D, in wild type (WT) mosquitoes, compare with PBS treatment, after fungal challenge (Bb), the relative mRNA level of DEFA is increase about 35-fold, and in TRAF4 heterozygous (TRAF4^{+/-}), the relative mRNA level of DEFA is increase about 1.5-fold. These results indicated that loss of only one copy of TRAF4 gene can significantly decrease the expression of DEFA. However, we can't confirm whether Toll pathway is completely blocked. The results of CECE are similar.

We added more qPCR results of selected AMPs in TRAF4 specific knockdown (iTRAF4) mosquitoes. The results are similar with TRAF4^{+/-} mosquitoes. The survival rate results (Fig S4A, 4B) also confirmed that loss of one copy of TRAF4 and TRAF4 specific RNAi have similar inhibitory effect on Toll pathway.

6. Regarding the writing:

- Curdlan should be introduced for non-mosquito audience.

We added the description of Curdlan in the manuscript.

- The use of "the" needs to be thoroughly revised.

We checked the whole article and revised it.

- Use either "...K63-linked polyubiquitination" or "xxx is K63-polyubiquitinated at...", rather than "K63-link polyubiquitinated"

We have checked the whole article, and changed "K63-link polyubiquitinated" into "K63-linked polyubiquitinated".

- I suggest precise terms to be used when making conclusions. e.g. line 214, use restricting rather than affecting; line 314, change "plays an important role in " to "is required for".

We changed "affecting" into "restricting" in the manuscript. And we also have changed "plays an important role in" into "is required for".

- Line 73, "cov unterattacks" is counterattacks?\

Yes. We checked the writing of the whole article carefully.

- Line 88, Ubiquitination, the ligation reaction catalyzed by ubiquitin ligase to removal by deubiquitinases (DUBs)..... Please re-write it.

- Line 93, "The process of ubiquitination requires the coordination of three kinds of

enzymes: E1 activating, E2 conjugating, and E3 ligating." Need to re-write.

We have re-written this paragraph. "Ubiquitination is a is considered one of the key signaling events in regulation of innate immune signaling pathways, particularly NF- κ B pathways in mammals and insects."

"The process of ubiquitination requires the coordination of three kinds of enzymes: ubiquitin-activating enzyme E1, ubiquitin-conjugating enzyme E2, and ubiquitin-ligase E3, which can be removed by deubiquitinases (DUBs)."

- Line 107, IMD does not stand for Immunodeficient, but rather it is abbreviation of immune deficiency.

We changed "Immunodeficient (Imd)" into "immune deficiency (Imd)".

- Line 122, OUT DUBs is OUT-domain containing DUBs?

"OTU DUBs" was changes to "OTU-domain containing DUBs".

- Line 242, "in contrast" should be "similarly".

We changed "In contrast" into "Similarly".

- Line 175, change to ".....genes involved in lipid metabolism increased after OTU7B knockdown (Fig. S2D), consistent to the notion that immune response is an energy-consuming process (26)".

We modified the sentence according to this suggestion.

- line 297, K63 mutation affects the enzymatic formation of polyubiquitin chains, please re-write.

We re-wrote the sentence to "The mutation of K63 in ubiquitin affects the formation of polyubiquitin chains in TRAF4."

- line 347, "...OTU7B inhibits TRAF4 from functioning in the Toll pathway by its deubiquitination" to "...OTU7B deubiquitinates TRAF4 thereby restricting its capacity to activate the Toll pathway".

We made changes according the reviewer's suggestion.

- Line 396, change to "PGRP and β GRP". What is β GRP?

We changed "PGRP, β GRP" into "PGRP and β GRP".

β GRP (β -1,3-glucan recognition protein), we added this explanation in the manuscript.

- Line 412, K63-linked polyubiquitination chains, change to "...polyubiquitin chains"

We changed “K63-linked polyubiquitination chains” into “K63-linked polyubiquitin chains”.

- Line 420, "In conclusion, we identified the mechanism by which the fungus *B. bassiana* affects the mosquito innate immunity (Fig. 7)." I believe this is a (nice) mechanism, but not the (only) mechanism.

We changed this sentence into “In conclusion, we identified a mechanism by which the fungus *B. bassiana* affects the mosquito innate immunity”.

Reviewer #2

Major comments:

1, In Fig1 A-C, the authors proved that, after fungal infection, the expression of OTU7B increases significantly. However, it would be clearer, if the author could give a detailed mechanism of how OTU7B is induced to express.

Thanks for the comments. We added new experiments to explain the mechanism of how OTU7B is induced to express. As shown in Fig 1C and Fig S1C, OTU7B was induced by mosquito Rel1A.

2, In figure 2A, the author only showed the translocation of Rel1 into nucleus at 48 h after fungal infection, why? Does the author have any results of other infection time?

As shown in Fig 1A, we can find that OTU7B is significantly induced at 48 h after fungal infection. Previous studies also showed that Toll pathway was activated at 48 h after fungal infection (Zou. et al, Immunity, 2010.). Therefore, we studied the translocation of Rel1 into nucleus at 48 h. Besides, we also test the translocation of Rel1 into nucleus at 24 and 72 h after fungal infection, no obvious difference was detected.

Minor comments:

1. The authors should check the English writing of the whole article carefully.

We checked the English writing and revised the manuscript carefully.

2. The author used "ubiquitin chains" in abstract and "polyubiquitin chains" in the whole manuscript. It is suggested to modify.

We changed the “ubiquitin chains” to “polyubiquitin chains” in abstract.

3. Line 103 and 107, "K63-linked" or "K63-link"? check it.

We changed “K63-link” into “K63-linked”.

4. Line 115, change "ovarian tumor" to "ovarian tumor (OTU)".

We changed "ovarian tumor" to "ovarian tumor (OTU)".

5. Line 141, change "Bb infection" to "B. bassiana infection".

We changed "Bb infection" to "*B. bassiana infection*".

6. Line 182, I think Fig 1C should be 1F, and line 187, Fig 1D should be 1G.

We rearranged the order of the figures and made corresponding marks.

7. In page 7, line 192, change "Ae. aegypti Rel1(AaRel1)" to "AaRel1".

"*Ae. aegypti* Rel1 (AaRel1), an orthologue of Dorsal, is a key molecule in the Toll immune pathway", this sentence appears twice in the article, here we deleted it.

Additional References:

Takahiro Tanji, Eun-Young Yun, and Y. Tony Ip. Heterodimers of NF- κ B transcription factors DIF and Relish regulate antimicrobial peptide genes in *Drosophila*. PNAS, 2010.8, 107, 14715-14720.

Zhen Zou, Sang Woon Shin, Kanwal S. Alvarez, Vladimir Kokoza, and Alexander S. Raikhel. Distinct Melanization Pathways in the Mosquito *Aedes aegypti*. Immunity, 2010.1, 32, 42-53.

November 29, 2022

Prof. Zhen Zou
Institute of Zoology, Chinese Academy of Sciences
Beijing
China

Re: Spectrum03123-22R1 (OTU7B modulates the mosquito immune response to *Beauveria bassiana* infection via deubiquitination of the Toll adaptor TRAF4)

Dear Prof. Zhen Zou:

Your manuscript has been accepted, and I am forwarding it to the ASM Journals Department for publication. You will be notified when your proofs are ready to be viewed.

Sincerely,

Chengshu Wang
Editor, Microbiology Spectrum
